# RAMBO-RL: Robust Adversarial Model-Based Offline Reinforcement Learning

**Marc Rigter, Bruno Lacerda, Nick Hawes**
Oxford Robotics Institute
University of Oxford
`{mrigter, bruno, nickh}@robots.ox.ac.uk`

## Abstract

Offline reinforcement learning (RL) aims to find performant policies from logged data without further environment interaction. Model-based algorithms, which learn a model of the environment from the dataset and perform conservative policy optimisation within that model, have emerged as a promising approach to this problem. In this work, we present Robust Adversarial Model-Based Offline RL (RAMBO), a novel approach to model-based offline RL. We formulate the problem as a two-player zero sum game against an adversarial environment model. The model is trained to minimise the value function while still accurately predicting the transitions in the dataset, forcing the policy to act conservatively in areas not covered by the dataset. To approximately solve the two-player game, we alternate between optimising the policy and adversarially optimising the model. The problem formulation that we address is theoretically grounded, resulting in a probably approximately correct (PAC) performance guarantee and a pessimistic value function which lower bounds the value function in the true environment. We evaluate our approach on widely studied offline RL benchmarks, and demonstrate that it outperforms existing state-of-the-art baselines.

## 1 Introduction

Reinforcement learning (RL) [61] has achieved state-of-the-art performance on many sequential decision-making problems [40, 45, 59]. However, the need for extensive exploration prohibits the application of RL to many real world domains where such exploration is costly or dangerous. Offline RL [33, 35] overcomes this limitation by learning policies from static, pre-recorded datasets.

Online RL algorithms perform poorly in the offline setting due to the distributional shift between the state-action pairs in the dataset and those taken by the learnt policy. Thus, an important aspect of offline RL is to introduce conservatism to prevent the learnt policy from executing state-action pairs which are out of distribution. Model-free offline RL algorithms [15, 23, 27, 29, 31, 72] train a policy from only the data present in the fixed dataset, and incorporate conservatism either into the value function or by directly constraining the policy.

On the other hand, model-based offline RL algorithms [76, 75, 25, 63, 39] use the dataset to learn a model of the environment, and train a policy using additional synthetic data generated from that model. By training on additional synthetic data, model-based algorithms can potentially generalise better to states not present in the dataset, or to solving new tasks. Previous approaches to model-based offline RL incorporate conservatism by estimating the uncertainty in the model and applying reward penalties for state-action pairs that have high uncertainty [76, 25]. However, uncertainty estimation can be unreliable for neural network models [75, 37]. Like recent work [75], we propose an approach for offline model-based RL which *does not require uncertainty estimation*.

36th Conference on Neural Information Processing Systems (NeurIPS 2022).

In this work we present Robust Adversarial Model-Based Offline (RAMBO) RL, a new algorithm for model-based offline RL. RAMBO incorporates conservatism by modifying the *transition dynamics* of the learnt Markov decision process (MDP) model in an adversarial manner. We formulate the problem of offline RL as a zero-sum game against an adversarial environment. To solve the resulting maximin optimisation problem, we alternate between optimising the agent and optimising the adversary in the style of Robust Adversarial RL (RARL) [50]. Unlike existing RARL approaches, our model-based approach forgoes the need to define and train an adversary policy, and instead only learns an adversarial model of the MDP. We train the agent policy with an actor-critic algorithm using synthetic data generated from the model in addition to data sampled from the dataset, similar to Dyna [60] and a number of recent methods [22, 76, 25, 75]. We update the environment model so that it reduces the value function for the agent policy, while still accurately predicting the transitions in the dataset. As a result, our approach introduces conservatism by generating *pessimistic synthetic transitions* for state-action pairs which are out-of-distribution. The theoretical formulation of offline RL that our algorithm addresses yields a PAC bound for the performance gap with respect to any policy covered by the dataset, and a pessimistic value function that lower bounds the value function in the true environment.

In summary, the main contributions of this work are:

- RAMBO, a novel and theoretically-grounded model-based offline RL algorithm which enforces conservatism by training an adversarial dynamics model.
- Adapting the Robust Adversarial RL approach to model-based offline RL by proposing a new formulation of RARL, where instead of defining and training an adversary policy, we directly train the model adversarially.

In our experiments we demonstrate that RAMBO outperforms current state-of-the-art algorithms on the D4RL benchmarks [14]. Furthermore, we provide ablation results which show that training the model adversarially is crucial to the strong performance of RAMBO.

## 2   Related Work

**Offline RL:** Offline RL addresses the problem of learning policies from fixed datasets, and has been applied to domains such as healthcare [42, 57], natural language processing [24, 23], and robotics [30, 38, 51]. Model-free offline RL algorithms do not require a learnt model. Approaches for model-free offline RL include importance sampling algorithms [36, 41], constraining the learnt policy to be similar to the behaviour policy [16, 28, 72, 23, 58, 15], incorporating conservatism into the value function during training [9, 27, 31, 73], using uncertainty quantification to generate more robust value estimates [1, 2, 29], or applying only a single iteration of policy iteration [7, 49]. In contrast, model-based approaches learn a model of the environment and generate synthetic data from that model [60] to optimise a policy using either planning [4] or RL algorithms [25, 76, 75]. By training a policy on additional synthetic data, model-based approaches have the potential for broader generalisation and for solving new tasks [6, 76]. A simple approach to ensuring conservatism is to constrain the policy to be similar to the behaviour policy in the same fashion as some model-free approaches [8, 39, 63]. Another approach is to apply reward penalties for executing state-action pairs with high uncertainty in the environment model [25, 74, 76]. However, this requires explicit uncertainty estimates which may be unreliable for neural network models [17, 37, 46, 75]. COMBO [75] obviates the need for uncertainty estimation in model-based offline RL by adapting model-free techniques [31] to regularise the value function for out-of-distribution samples. Like COMBO, our approach does not require uncertainty estimation.

Most approaches to model-based offline RL use maximum likelihood estimates (MLE) of the MDP trained using standard supervised learning [4, 39, 63, 76, 75]. However, other methods have been proposed to learn models which are more suitable for offline policy optimisation. One approach is to reweight the loss function to ensure the model is accurate under the state-action distribution generated by the policy [34, 53, 20]. In contrast, our approach produces pessimistic synthetic transitions when out-of-distribution.

Most related to our work is a recent paper [66] which introduces the maximin formulation of offline RL that we address. This existing work motivates our approach theoretically by showing that the problem formulation obtains probably approximately correct (PAC) guarantees. However, [66]

only addresses the theoretical aspects of the problem formulation and does not propose a practical algorithm. In this work, we propose a practical RL algorithm to solve the maximin formulation of model-based offline RL.

**Robust RL:** Algorithms for Robust MDPs [5, 12, 21, 43, 54, 64, 70] find the policy with the best worst-case performance over a set of possible MDPs. Typically, it is assumed that the uncertainty set of MDPs is specified a priori. To eliminate the need to specify the set of possible MDPs, model-free approaches to Robust MDPs [68, 55] instead assume that samples can be drawn from a misspecified MDP which is similar to the true MDP. As our work addresses offline RL, we assume that we have a fixed dataset from the true MDP.

Our approach is conceptually similar to Robust Adversarial RL (RARL) [50], a method proposed to improve the robustness of RL policies in the *online* setting. RARL is posed as a two-player zero-sum game where the agent plays against an adversary which perturbs the environment. Formulations of model-free RARL differ in how they define the action space of the adversary. Options include allowing the adversary to apply perturbation forces to the simulator [50], add noise to the agent's actions [65], or periodically take over control [48]. A model-based approach to RARL is proposed in [13], which learns an optimistic and pessimistic model to encourage online exploration. However, this existing approach requires uncertainty estimation as well as an adversarial policy to be learnt in *addition to the model*. Our work follows the paradigm of RARL and alternates between agent and adversarial updates in a maximin formulation. We adapt model-based RARL to the offline setting and propose an alternative formulation: we eliminate the need to learn an adversary policy and instead *directly modify the MDP model adversarially*.

## 3 Preliminaries

**MDPs and Offline RL:** An MDP is defined by the tuple, $M = (S, A, T, R, \mu_0, \gamma)$. $S$ and $A$ denote the state and action spaces respectively, $R(s, a)$ is the reward function, $T(s'|s, a)$ is the transition function, $\mu_0$ is the initial state distribution, and $\gamma \in (0, 1)$ is the discount factor. In this work we consider Markovian policies, $\pi \in \Pi$, which map from each state to a distribution over actions. We denote the (improper) discounted state visitation distribution of a policy by $d_M^\pi(s) := \sum_{t=0}^{\infty} \gamma^t \Pr(s_t = s|\pi, M)$, where $\Pr(s_t = s|\pi, M)$ is the probability of reaching state $s$ at time $t$ by executing policy $\pi$ in $M$. The improper state-action visitation distribution is $d_M^\pi(s, a) = \pi(a|s) \cdot d_M^\pi(s)$. We also denote the normalised state-action visitation distribution by $\tilde{d}_M^\pi(s, a) = (1 - \gamma) \cdot d_M^\pi(s, a)$.

The value function, $V_M^\pi(s)$, represents the expected discounted return from executing $\pi$ from state $s$ in $M$: $V_M^\pi(s) = \mathbb{E}_{\pi, M}\left[\sum_{t=0}^{\infty} \gamma^t R(s_t, a_t)\right]$. We write $V_M^\pi$ to indicate the value function under the initial state distribution, i.e. $V_M^\pi = \sum_{s \in S} \mu_0(s) V_M^\pi(s)$. The standard objective for MDPs is to find the policy which maximises $V_M^\pi$. The state-action value function, $Q_M^\pi(s, a)$, is the expected discounted cumulative reward from taking action $a$ at state $s$ and then executing $\pi$ thereafter.

In offline RL we only have access to a fixed dataset of transitions from the MDP, $\mathcal{D} = \{(s_i, a_i, r_i, s_i')\}_{i=1}^{|\mathcal{D}|}$. The goal of offline RL is to find the best possible policy using the fixed dataset.

**Model-Based Offline RL Algorithms:** Model-based approaches to offline RL use a model of the MDP to help train a policy. The dataset is used to learn a dynamics model, $\widehat{T}$, which is typically trained via maximum likelihood estimation: $\min_{\widehat{T}} \mathbb{E}_{(s,a,s') \sim \mathcal{D}}\left[-\log \widehat{T}(s'|s, a)\right]$. A model of the reward function, $\widehat{R}(s, a)$, can also be learnt if it is unknown. The estimated MDP, $\widehat{M} = (S, A, \widehat{T}, \widehat{R}, \mu_0, \gamma)$, has the same state and action space as the true MDP but uses the learnt transition and reward functions. Thereafter, any planning or RL algorithm can be used to recover optimal policy in the learnt model, $\widehat{\pi} = \arg\max_{\pi \in \Pi} V_{\widehat{M}}^\pi$.

Unfortunately, directly applying this approach to the offline RL setting does not perform well due to distributional shift. In particular, if the dataset does not cover the entire state-action space, the model will inevitably be inaccurate for some state-action pairs. Thus, naive policy optimisation on a learnt model in the offline setting can result in *model exploitation* [22, 32, 53]. To mitigate this issue, we propose the novel approach of enforcing conservatism by adversarially modifying the transition dynamics of $\widehat{M}$.

In line with existing works [76, 75, 8], we use model-based policy optimisation (MBPO) [22] to learn the optimal policy for $\widehat{M}$. MBPO utilises a standard actor-critic RL algorithm. However, the value function is trained using an augmented dataset $\mathcal{D} \cup \mathcal{D}_{\widehat{M}}$, where $\mathcal{D}_{\widehat{M}}$ is synthetic data generated by simulating rollouts in the learnt model. To generate the synthetic data, MBPO performs $k$-step rollouts in $\widehat{M}$ starting from states $s \in \mathcal{D}$, and adds this data to $\mathcal{D}_{\widehat{M}}$. To train the policy, minibatches of data are drawn from $\mathcal{D} \cup \mathcal{D}_{\widehat{M}}$, where each datapoint is sampled from the real data, $\mathcal{D}$, with probability $f$, and from $\mathcal{D}_{\widehat{M}}$ with probability $1 - f$.

**Robust Adversarial Reinforcement Learning:** RARL addresses the problem of finding a robust agent policy, $\pi$, in the online RL setting by posing the problem as a two-player zero sum game against adversary policy, $\bar{\pi}$:

$$\pi = \arg\max_{\pi \in \Pi} \min_{\bar{\pi} \in \bar{\Pi}} V_M^{\pi, \bar{\pi}} \tag{1}$$

where $V_M^{\pi, \bar{\pi}}$ is the expected value from executing $\pi$ and $\bar{\pi}$ in environment $M$. Different approaches define the action space for $\bar{\pi}$ in different ways, as discussed in Section 2. For a scalable approximation to the optimisation problem in Equation 1, algorithms for RARL alternate between applying steps of stochastic gradient ascent to the agent's policy to increase the expected value, and stochastic gradient descent to the adversary's policy to decrease the expected value. In our work, we follow the RARL paradigm of alternating between agent and adversarial updates and adapt it to the model-based offline setting. Instead of defining a separate adversary policy, we treat the *model itself* as the policy to be adversarially trained.

# 4 Problem Formulation

For the sake of generality, we assume that both the transition function and reward function are unknown. Hereafter, we write $\widehat{T}$ to denote both the learnt dynamics and reward function, where $\widehat{T}(s', r | s, a)$ is the probability of receiving reward $r$ and transitioning to $s'$ after executing $(s, a)$. We address the maximin formulation of offline RL recently proposed by [66]:

**Problem 1.** *For some dataset, $\mathcal{D}$, and some fixed constant $\xi > 0$, find the policy $\pi$ defined by*

$$\pi = \arg\max_{\pi \in \Pi} \min_{\widehat{T} \in \mathcal{M}_{\mathcal{D}}} V_{\widehat{T}}^{\pi}, \text{ where} \tag{2}$$

$$\mathcal{M}_{\mathcal{D}} = \left\{ \widehat{T} \mid \mathbb{E}_{\mathcal{D}} \left[ \mathrm{TV}(\widehat{T}_{\mathrm{MLE}}(\cdot | s, a), \widehat{T}(\cdot | s, a))^2 \right] \leq \xi \right\}, \tag{3}$$

*where $\mathrm{TV}(P_1, P_2)$ is the total variation distance between distributions $P_1$ and $P_2$, and $\widehat{T}_{\mathrm{MLE}}$ denotes the maximum likelihood estimate of the MDP given the offline dataset, $\mathcal{D}$.*

Thus, the set defined in Equation 3 contains MDPs which are similar to the maximum likelihood estimate under state-action pairs in $\mathcal{D}$. However, because the expectation in Equation 3 is taken under $\mathcal{D}$, there is no restriction on $\widehat{T}$ for regions of the state-action space not covered by $\mathcal{D}$. We present a brief overview of the theoretical guarantees from [66] in the following subsection.

**Remark 1.** Note that Problem 1 differs from the pessimistic MDP formulations introduced by MOPO [76] and MOReL [25]. Problem 1 considers the worst-case transition dynamics, while the pessimistic MDPs constructed by MOPO and MOReL only modify the reward function by applying reward penalties for state-action pairs with high uncertainty.

## 4.1 Theoretical Motivation

The theoretical analysis from [66] shows that solving Problem 1 outputs a policy that with high probability is approximately as good as any policy with a state-action distribution that is covered by the dataset. This is formally stated in the following theorem.

**Theorem 1** (PAC guarantee from [66], Theorem 5)**.** *Denote the true MDP transition function by $T$, and let $\mathcal{M}$ denote a hypothesis class of MDP models such that $T \in \mathcal{M}$. Let $\pi$ denote the solution to Problem 1 for dataset $\mathcal{D}$. Then with probability $1 - \delta$ for any policy, $\pi^* \in \Pi$, we have*

$$V_T^{\pi^*} - V_T^{\pi} \leq (1 - \gamma)^{-2} c_1 \sqrt{C_{\pi^*}} \sqrt{G_{\mathcal{M}_1} + G_{\mathcal{M}_2} + \xi_n^2 + \frac{\ln(c/\delta)}{|\mathcal{D}|}}, \text{ where}$$

$$C_{\pi^*} = \max_{T' \in \mathcal{M}} \frac{\mathbb{E}_{(s,a) \sim \tilde{d}_T^{\pi^*}}\left[\text{TV}(T'(\cdot|s,a), T(\cdot|s,a))^2\right]}{\mathbb{E}_{(s,a) \sim \rho}\left[\text{TV}(T'(\cdot|s,a), T(\cdot|s,a))^2\right]},$$

*where $\rho$ is the state-action distribution from which $\mathcal{D}$ was sampled, and $c$ and $c_1$ are universal constants. We refer the reader to Appendix A of [66] for the definitions of $G_{\mathcal{M}_1}$, $G_{\mathcal{M}_2}$, and $\xi_n$.*

The quantity $C_{\pi^*}$ is upper bounded by the maximum density ratio between the comparator policy, $\pi^*$, and the offline distribution, i.e. $C_{\pi^*} \leq \max_{(s,a)} \tilde{d}_T^{\pi^*}(s,a)/\rho(s,a)$. It represents the discrepancy between the distribution of data in the dataset compared to the visitation distribution of policy $\pi^*$. Theorem 1 shows that if we find a policy by solving Problem 1, the performance gap of that policy is bounded with respect to any other policy $\pi^*$ that has a state-action distribution which is covered by the dataset.

Furthermore, the value function under the worst-case model in the set defined by Problem 1 is a lower bound on the value function in the true environment, as stated by Proposition 1.

**Proposition 1** (Pessimistic value function). *Let $T$ denote the true transition function for some MDP, and let $\mathcal{M}_{\mathcal{D}}$ be the set of MDP models defined in Equation 3. Then for any policy $\pi$, with probability $1 - \delta$ we have that*

$$\min_{\widehat{T} \in \mathcal{M}_{\mathcal{D}}} V_{\widehat{T}}^\pi \leq V_T^\pi.$$

Proposition 1 follows from the fact that $T \in \mathcal{M}_{\mathcal{D}}$ with high probability, which is proven in [66] (Appendix E.2). Proposition 1 shows that we can expect the performance of any policy in the true MDP to be at least as good as the value in the worst-case model defined in Problem 1.

While [66] provides the theoretical motivation for solving Problem 1, it does not propose a practical algorithm. In this work, we focus on developing a practical approach to solving Problem 1.

# 5 RAMBO-RL

In this section, we present Robust Adversarial Model-Based Offline RL (RAMBO), our algorithm for solving Problem 1. The main difficulty with solving Problem 1 is that it is unclear how to find the worst-case MDP in the set defined in Equation 3. To arrive at a scalable solution, we propose a novel approach which is in the spirit of RARL. We alternate between optimising the agent policy to increase the expected value, and adversarially optimising the model to decrease the expected value. In this section, we first describe how we compute the gradient to adversarially train the model. Then, we discuss how to ensure that the model remains approximately within the constraint set defined in Problem 1. Finally, we present our overall algorithm.

## 5.1 Model Gradient

We propose a policy gradient-inspired approach to adversarially optimise the model. Typically, policy gradient algorithms are used to modify the distribution over actions taken by a policy at each state [62, 71]. In contrast, the update that we propose modifies the likelihood of the successor states and rewards in the MDP model.

We assume that the MDP model is defined by parameters, $\phi$, and we write $\widehat{T}_\phi$ to indicate this. We denote by $V_\phi^\pi$ the value function for policy $\pi$ in model $\widehat{T}_\phi$. To approximately find $\min_{\widehat{T}_\phi \in \mathcal{M}_{\mathcal{D}}} V_\phi^\pi$ as required by Problem 1 via gradient descent, we wish to compute the gradient of the model parameters that reduces the value of the policy within the model, i.e. $\nabla_\phi V_\phi^\pi$.

**Proposition 2** (Model Gradient). *Let $\phi$ denote the parameters of a parametric MDP model $\widehat{T}_\phi$, and let $V_\phi^\pi$ denote the value function for policy $\pi$ in $\widehat{T}_\phi$. Then:*

$$\nabla_\phi V_\phi^\pi = \mathbb{E}_{s \sim d_\phi^\pi, a \sim \pi, (s',r) \sim \widehat{T}_\phi}\left[(r + \gamma V_\phi^\pi(s')) \cdot \nabla_\phi \log \widehat{T}_\phi(s',r|s,a)\right] \quad (4)$$

The proof of Proposition 2 is given in Appendix A. We can subtract the baseline $Q_\phi^\pi(s,a)$ without biasing the gradient estimate (see Appendix A.1 for details):

$$\nabla_\phi V_\phi^\pi = \mathbb{E}_{s \sim d_\phi^\pi, a \sim \pi, (s',r) \sim \widehat{T}_\phi}\left[(r + \gamma V_\phi^\pi(s') - Q_\phi^\pi(s,a)) \cdot \nabla_\phi \log \widehat{T}_\phi(s',r|s,a)\right] \quad (5)$$

The Model Gradient differs from the standard policy gradient in that a) it is used to update the likelihood of successor states in the model, rather than actions in a policy, and b) the "advantage" term $r + \gamma V_\phi^\pi(s') - Q_\phi^\pi(s,a)$ compares the utility of receiving reward $r$ and transitioning to $s'$ to the expected value of state action pair $(s,a)$. In contrast, the standard advantage term, $Q(s,a) - V(s)$ [56], compares the value of executing state-action pair $(s,a)$ to the value at $s$. To estimate $V_\phi^\pi$ and $Q_\phi^\pi$, we use the critic learnt by the actor-critic algorithm used for policy optimisation. Thus, we use the critic *both* for training the policy and adversarially training the model.

**Remark 2.** The Model Gradient can be thought of as a specific instantiation of the policy gradient if we view $\widehat{T}_\phi$ as a adversarial "policy" on an augmented MDP, $M^+$, i.e. $\widehat{T}_\phi : S^+ \to \mathrm{Dist}(A^+)$. The augmented state space, $S^+ = S \times A$ includes the state in the original MDP augmented by the action taken by the agent. The augmented action space consists of the reward applied and the successor state, $A^+ = S \times [R_{\min}, R_{\max}]$. Thus, we can think of this approach as an instantiation of RARL in which the adversary policy ($\bar{\pi}$ in Equation 1) that we train is the *model itself*.

## 5.2 Adversarial Model Training

If we were to update the model using Equation 5 alone, this would allow the model to be modified arbitrarily such that the value function in the model is reduced. However, the set of plausible MDPs given by Equation 3 states that over the dataset, $\mathcal{D}$, the model $\widehat{T}_\phi$ should be close to the maximum likelihood estimate, $\widehat{T}_{\mathrm{MLE}}$. Specifically, in the inner optimisation of Problem 1 we wish to find a solution to the constrained optimisation problem

$$\min_{\widehat{T}_\phi} V_\phi^\pi, \quad s.t. \ \mathbb{E}_{\mathcal{D}}\big[\mathrm{TV}(\widehat{T}_{\mathrm{MLE}}(\cdot|s,a), \widehat{T}_\phi(\cdot|s,a))^2\big] \leq \xi. \tag{6}$$

The Lagrangian relaxation leads to the unconstrained problem

$$\max_{\lambda \geq 0} \min_{\widehat{T}_\phi} \Big( L(\widehat{T}, \lambda) := V_\phi^\pi + \lambda\big(\mathbb{E}_{\mathcal{D}}\big[\mathrm{TV}(\widehat{T}_{\mathrm{MLE}}(\cdot|s,a), \widehat{T}_\phi(\cdot|s,a))^2\big] - \xi\big)\Big), \tag{7}$$

where $\lambda$ is the Lagrange multiplier. Rather than optimising the Lagrange multiplier, we find that in practice fixing $\lambda$ to apply a constant weighting between the two terms works well with minimal tuning. To facilitate easier tuning of the learning rate, in our implementation we apply the weighting constant to the value function term rather than the model term, which is equivalent up to a scaling factor. This leads to

$$\min_{\widehat{T}_\phi} \Big( \lambda V_\phi^\pi + \mathbb{E}_{\mathcal{D}}\big[\mathrm{TV}(\widehat{T}_{\mathrm{MLE}}(\cdot|s,a), \widehat{T}_\phi(\cdot|s,a))^2\big]\Big). \tag{8}$$

We aim to make our algorithm efficient and simple to implement. Therefore, rather than minimising the TV distance between the model and MLE model as prescribed by Equation 8, we directly optimise the standard MLE loss. This leads to the final loss function:

$$\mathcal{L}_\phi = \lambda V_\phi^\pi - \mathbb{E}_{(s,a,r,s') \sim \mathcal{D}}\big[\log \widehat{T}_\phi(s', r|s,a)\big]. \tag{9}$$

Thus, the loss function for the model in Equation 9 simply adds the adversarial term to the standard MLE loss. By minimising the loss function in Equation 9, the model is trained to a) predict the transitions within the dataset, and b) reduce the value function of the policy, with $\lambda$ determining the tradeoff between these two objectives. Choosing $\lambda$ to be small ensures that the MLE term dominates for transitions within $\mathcal{D}$, ensuring that the model fits the dataset accurately. Because the MLE term is only computed over $\mathcal{D}$, the adversarial term dominates outside of the dataset meaning that the model is modified adversarially for transitions outside of the dataset.

To estimate the gradient of the loss function in Equation 9 for stochastic gradient descent, we sample a minibatch of transitions from $\mathcal{D}$ to estimate the MLE term. The gradient for the value function term is computed using the Model Gradient. The transitions used to compute the Model Gradient term must be sampled under the current policy and model (Equation 5). Therefore, to estimate the Model Gradient term, we generate a minibatch of transitions by simulating the current policy in $\widehat{T}_\phi$.

Like previous works [10, 76, 75, 8], we represent the dynamics model using an ensemble of neural networks. Each neural network produces a Gaussian distribution over the next state and reward:

$\widehat{T}_\phi(s', r|s, a) = \mathcal{N}(\mu_\phi(s, a), \Sigma_\phi(s, a))$. A visualisation of the result of adversarially training the dynamics model can be found in Appendix C.3.

**Normalisation** The composite loss function in Equation 9 comprises two terms which may have different magnitudes across domains depending on the scale of the states and rewards. To enable easier tuning of the adversarial loss weighting, $\lambda$, across different domains we perform the following normalisation procedure. Prior to training, we normalise the states in $\mathcal{D}$ in the manner proposed in [15], by subtracting the mean and dividing by the standard deviation of each state dimension in the dataset. Additionally, when computing the gradient in Equation 5 we normalise the advantage terms, $r + \gamma V_\phi^\pi(s') - Q_\phi^\pi(s, a)$, according to the mean and standard deviation across each minibatch. Advantage normalisation is common practice in policy gradient RL implementations [3, 52].

### 5.3 Algorithm

We are now ready to present our overall approach in Algorithm 1. The first step of RAMBO is to pretrain the environment dynamics model using standard MLE (Line 1). Thereafter, the algorithm follows the format of RARL. At each iteration, we apply gradient updates to the agent to increase the expected value, followed by gradient updates to the model to decrease the expected value.

Prior to each agent update, we generate synthetic $k$-step rollouts starting from states in $\mathcal{D}$ by simulating rollouts in the current MDP model $\widehat{T}_\phi$. This data is added to the synthetic dataset $\mathcal{D}_{\widehat{T}_\phi}$ (Line 3). Following previous approaches [76, 75, 22] we only store data from recent iterations in $\mathcal{D}_{\widehat{T}_\phi}$. The agent's policy and value functions are trained with an off-policy actor-critic algorithm using samples from $\mathcal{D} \cup \mathcal{D}_{\widehat{T}_\phi}$ (Line 4). In our implementation, we use soft actor-critic (SAC) [19] for agent training. To update the model to minimise the loss in Equation 9 (Line 5), we sample data from $\mathcal{D}$ to estimate the gradient for the MLE component. To compute the adversarial component we generate samples by simulating the current policy and model, and utilise the value function learnt by the agent to compute the gradient according to Equation 5.

---

**Algorithm 1** RAMBO-RL

---

**Require:** Normalised dataset, $\mathcal{D}$;
 1: $\widehat{T}_\phi \leftarrow$ MLE dynamics model.
 2: **for** $i = 1, 2, \ldots, n_{\text{iter}}$ **do**
 3:     Generate synthetic $k$-step rollouts. Add transition data to $\mathcal{D}_{\widehat{T}_\phi}$.
 4:     *Agent update*: Update $\pi$ and $Q_\phi^\pi$ with an actor critic algorithm, using samples from $\mathcal{D} \cup \mathcal{D}_{\widehat{T}_\phi}$.
 5:     *Adversarial model update*: Update $\widehat{T}_\phi$ according to Eq. 9, using samples from $\mathcal{D}$ for the MLE component, and the current critic $Q_\phi^\pi$ and synthetic data sampled from $\pi$ and $\widehat{T}_\phi$ for the adversarial component.

---

## 6 Experiments

In our experiments, we aim to: a) evaluate how well RAMBO performs compared to state-of-the-art baselines, b) examine whether RAMBO can be tuned offline, c) determine the impact of adversarial training on the performance of the algorithm, and d) investigate the difference between RAMBO and COMBO, the most similar prior algorithm. The code for our experiments is available at github.com/marc-rigter/rambo. We evaluate our approach on the following domains.

**MuJoCo** There are three different environments representing different robots (*HalfCheetah, Hopper, Walker2D*), each with 4 datasets (*Random, Medium, Medium-Replay, Medium-Expert*). *Random* contains transitions collected by a random policy. *Medium* contains transitions collected by an early-stopped SAC policy. *Medium-Replay* consists of the replay buffer generated while training the *Medium* policy. The *Medium-Expert* dataset contains a mixture of suboptimal and expert data.

**AntMaze** The agent controls a robot and navigates to reach a goal, receiving a sparse reward only if the goal is reached. There are three different layouts of maze (*Umaze, Medium, Large*), and different dataset types (*Fixed, Play, Diverse*) which differ in terms of the variety of start and goal locations used to collect the dataset. The MuJoCo and AntMaze benchmarks are from D4RL [14].

**Hyperparameter Details**   The base hyperparameters that we use for RAMBO mostly follow those used in SAC [19] and COMBO [75]. We find that the performance of RAMBO is sensitive to the choice of rollout length, $k$, consistent with findings in previous works [22, 37]. The other critical parameter for RAMBO is the choice of the adversarial weighting, $\lambda$.

For each dataset, we choose the rollout length and the adversarial weighting from one of three possible configurations: $(k, \lambda) \in \{(2, 3e\text{-}4), (5, 3e\text{-}4), (5, 0)\}$. We included $(k, \lambda) = (5, 0)$ as we found that an adversarial weighting of 0 worked well for some datasets. For the MuJoCo datasets we performed model rollouts using the current policy, and initialised the policy using behaviour cloning (BC) which is a common practice in offline RL [25, 74]. Ablation results in Appendix C.4 indicate that the BC initialisation results in a small improvement. For the AntMaze datasets we used a random rollout policy and a randomly initialised policy as we found that this performed better. Further details about the hyperparameters are in Appendix B.

**Evaluation**   We present two different evaluations of our approach: RAMBO and RAMBO[OFF]. For RAMBO, we ran each of the three hyperparameter configurations for five seeds each, and report the best performance across the three configurations. Thus, our evaluation of RAMBO utilises limited online tuning which is the most common practice among existing model-based offline RL algorithms [25, 37, 39, 76]. The performance obtained for each of the hyperparameter configurations is included in Appendix C.2.

Offline hyperparameter selection is an important topic in offline RL [47, 77]. Therefore, we present additional results for RAMBO[OFF] where we select between the three choices of hyperparameters offline using a simple heuristic (details in Appendix B.5) based on the magnitude and stability of the $Q$-values during offline training. We first select the hyperparameters using the heuristic, and then rerun each dataset for 5 seeds to generate the results for RAMBO[OFF].

**Baselines**   We compare RAMBO against state-of-the-art model-based (COMBO [75], RepB-SDE [34], MOReL [25], and MOPO [76]) and model-free (CQL [31], IQL [28], and TD3+BC [15]) offline RL algorithms. We provide results for all algorithms for the MuJoCo-v2 D4RL datasets and the AntMaze-v0 datasets (details in Appendix B.7).

## 6.1   Results

**D4RL Performance**   The results in Table 1 show that RAMBO achieves the best total score for the MuJoCo locomotion domains, outperforming existing state-of-the-art methods. Furthermore, RAMBO obtains the best overall score on both the Medium and Medium-Replay dataset types. For the random datasets, RAMBO is outperformed only by MOReL. This shows that RAMBO performs very well for datasets that are either noisy or consist of suboptimal data.

For the Medium-Expert datasets, RAMBO is outperformed by most of the baseline algorithms, suggesting that RAMBO is less suitable for high-quality datasets. However, for the Medium-Expert datasets, simpler approaches such as performing behaviour cloning on the best 10% of trajectories can be used to achieve stronger performance than offline RL methods [28]. Therefore, the suboptimal performance of RAMBO on the Medium-Expert datasets is less of a concern, as applying offline RL algorithms may not be the most suitable approach for these high-quality datasets.

For AntMaze, the model-based algorithms perform considerably less well than the model-free approaches. Unlike the other model-based approaches, RAMBO at least scores greater than zero for most of the datasets. Our results echo previous findings that model-based approaches struggle to perform well in the AntMaze domains, potentially because model-based algorithms are too aggressive and collide with walls [67]. Recent work [67] showed that *reverse* rollouts can lead to stronger performance for model-based methods in these domains. In future work, we wish to investigate whether combining RAMBO with reverse rollouts improves the performance for AntMaze.

**Offline Tuning**   In Table 1, we also present results for RAMBO[OFF] where the final hyperparameters are chosen using the heuristic described in Appendix B.5. We see that there is a slight degradation in the performance relative to RAMBO, which uses online tuning. However, RAMBO[OFF] still achieves comparable performance to the best existing approaches on the MuJoCo datasets. This suggests that suitable hyperparameters for RAMBO can reliably be chosen using our offline heuristic.

Table 1: Results for the D4RL benchmark using the normalisation procedure proposed by [14]. We report the normalised performance during the last 10 iterations of training averaged over 5 seeds. $\pm$ captures the standard deviation over seeds. Highlighted numbers indicate results within 2% of the most performant algorithm. * indicates the total without random datasets.

| | | Ours | | Model-based baselines | | | | Model-free baselines | | | |
|---|---|---|---|---|---|---|---|---|---|---|---|
| | | **RAMBO** | **RAMBO$^{OFF}$** | **RepB-SDE** | **COMBO** | **MOPO** | **MOReL** | **CQL** | **IQL** | **TD3+BC** | **BC** |
| Random | HalfCheetah | $40.0 \pm 2.3$ | $33.5 \pm 2.6$ | 32.9 | 38.8 | 35.4 | 25.6 | 19.6 | - | 11.0 | 2.1 |
| | Hopper | $21.6 \pm 8.0$ | $15.5 \pm 9.4$ | 8.6 | 17.9 | 4.1 | 53.6 | 6.7 | - | 8.5 | 9.8 |
| | Walker2D | $11.5 \pm 10.5$ | $0.2 \pm 0.6$ | 21.1 | 7.0 | 4.2 | 37.3 | 2.4 | - | 1.6 | 1.6 |
| Medium | HalfCheetah | $77.6 \pm 1.5$ | $71.0 \pm 3.0$ | 49.1 | 54.2 | 69.5 | 42.1 | 49.0 | 47.4 | 48.3 | 36.1 |
| | Hopper | $92.8 \pm 6.0$ | $91.2 \pm 16.3$ | 34.0 | 94.9 | 48.0 | 95.4 | 66.6 | 66.3 | 59.3 | 29.0 |
| | Walker2D | $86.9 \pm 2.7$ | $89.1 \pm 2.7$ | 72.1 | 75.5 | -0.2 | 77.8 | 83.8 | 78.3 | 83.7 | 6.6 |
| Medium Replay | HalfCheetah | $68.9 \pm 2.3$ | $67.0 \pm 1.5$ | 57.5 | 55.1 | 68.2 | 40.2 | 47.1 | 44.2 | 44.6 | 38.4 |
| | Hopper | $96.6 \pm 7.0$ | $97.6 \pm 3.4$ | 62.2 | 73.1 | 39.1 | 93.6 | 97.0 | 94.7 | 60.9 | 11.8 |
| | Walker2D | $85.0 \pm 15.0$ | $88.5 \pm 4.0$ | 49.8 | 56.0 | 69.4 | 49.8 | 88.2 | 73.9 | 81.8 | 11.3 |
| Medium Expert | HalfCheetah | $93.7 \pm 10.5$ | $79.3 \pm 2.9$ | 55.4 | 90.0 | 72.7 | 53.3 | 90.8 | 86.7 | 90.7 | 35.8 |
| | Hopper | $83.3 \pm 9.1$ | $89.5 \pm 11.1$ | 82.6 | 111.1 | 3.3 | 108.7 | 106.8 | 91.5 | 98.0 | 111.9 |
| | Walker2D | $68.3 \pm 20.6$ | $63.1 \pm 31.3$ | 88.8 | 96.1 | -0.3 | 95.6 | 109.4 | 109.6 | 110.1 | 6.4 |
| **MuJoCo-v2 Total:** | | $826.2 \pm 33.8$ | $785.5 \pm 40.4$ | 614.1 | 769.7 | 413.4 | 773.0 | 767.4 | 692.6* | 698.5 | 300.8 |
| AntMaze | Umaze | $25.0 \pm 12.0$ | $23.8 \pm 15.0$ | 0.0 | 80.3 | 0.0 | 0.0 | 74.0 | 87.5 | 78.6 | 65.0 |
| | Medium-Play | $16.4 \pm 17.9$ | $5.6 \pm 10.9$ | 0.0 | 0.0 | 0.0 | 0.0 | 61.2 | 71.2 | 3.0 | 0.0 |
| | Large-Play | $0.0 \pm 0.0$ | $0.0 \pm 0.0$ | 0.0 | 0.0 | 0.0 | 0.0 | 15.8 | 39.6 | 0.0 | 0.0 |
| | Umaze-Diverse | $0.0 \pm 0.0$ | $0.0 \pm 0.0$ | 0.0 | 57.3 | 0.0 | 0.0 | 84.0 | 62.2 | 71.4 | 55.0 |
| | Medium-Diverse | $23.2 \pm 14.2$ | $8.4 \pm 9.9$ | 0.0 | 0.0 | 0.0 | 0.0 | 53.7 | 70.0 | 10.6 | 0.0 |
| | Large-Diverse | $2.4 \pm 3.3$ | $0.0 \pm 0.0$ | 0.0 | 0.0 | 0.0 | 0.0 | 14.9 | 47.5 | 0.2 | 0.0 |
| **AntMaze-v0 Total:** | | $67.0 \pm 14.9$ | $37.8 \pm 12.4$ | 0.0 | 137.6 | 0.0 | 0.0 | 303.6 | 378.0 | 163.8 | 120.0 |

Table 2: Ablation of the adversarial updates for RAMBO. These results use the same rollout length for each dataset as RAMBO but with no adversarial updates (i.e. $\lambda = 0$). The scores are averaged over 5 seeds.

| **RAMBO (No Adversarial Training)** | |
|---|---|
| **MuJoCo-v2 Total:** $694.4 \pm 56.5$ | **AntMaze-v0 Total:** $45.8 \pm 21.8$ |

Table 3: Comparison between RAMBO and COMBO for the Single Transition Example. We use 20 seeds and $\pm$ captures the standard deviation over seeds. RAMBO outperforms COMBO ($p = 0.005$).

| **RAMBO:** $1.49 \pm 0.05$ | **COMBO:** $1.41 \pm 0.12$ |
|---|---|

**Ablation of Adversarial Training**    In Table 2 we present results for RAMBO with no adversarial updates. These results demonstrate that overall performance degrades if the adversarial training is removed. This parallels previous findings that mitigating the issue of model exploitation is crucial to obtaining strong performance in model-based offline RL. Interestingly however, for some specific datasets we obtain the best performance with no adversarial training (Appendix C.2). This suggests that a potential direction for future work could be trying to identify which types of problems do not require regularisation for a successful policy to be trained offline with model-based RL.

**Comparison to COMBO**    We focus especially on comparing our approach to COMBO, as it is the most similar existing algorithm. We compare RAMBO and COMBO on the Single Transition toy example which is described in detail in Appendix C.1. This domain has a one-dimensional state and action space, and several distinct regions of the action space are covered by the dataset. We use this domain to investigate whether the policies optimised by RAMBO and COMBO tend to become stuck in local optima.

Table 3 compares the performance of RAMBO and COMBO on the Single Transition Example. Further analysis in Appendix C.1 shows that for this problem, the pessimistic value function updates used by COMBO create local maxima in the $Q$-function which are present throughout training. Policy optimisation can become stuck in these local maxima. On the other hand, the value function produced by RAMBO is initially optimistic, and pessimism is introduced into the value function *gradually* as the transition function is modified adversarially. Adversarial modification of the transition function is visualised in Appendix C.3. As a result of this gradual introduction of pessimism, we observe that the

policy produced by RAMBO is less likely to become stuck in poor local maxima, and better overall performance is obtained in Table 3. This observation may help to explain why RAMBO is able to achieve consistently strong performance relative to existing algorithms in the MuJoCo domains. Gradually increasing the level of pessimism could be a useful modification for existing offline RL algorithms to be investigated in future work.

## 7 Conclusion and Future Directions

RAMBO is a promising new approach to offline RL which imposes conservatism by adversarially modifying the transition dynamics of a learnt model. Our approach is theoretically justified, and achieves state-of-the-art performance on standard benchmarks.

There are a number of possible extensions to RAMBO, some of which we have already discussed. In addition, we would like to apply RAMBO to image-space domains by using deep latent variable models to compress the state space [18, 51] and adversarially perturbing the transition dynamics in the latent space representation. Another direction that we would like to investigate is the use of adversarially trained models to aid interpretability in deep RL [44] by generating imagined worst-case trajectories. Finally, we wish to investigate applying the ideas developed in this work to the online RL setting to improve robustness.

## Acknowledgements

This work was supported by a Programme Grant from the Engineering and Physical Sciences Research Council (EP/V000748/1), the Clarendon Fund at the University of Oxford, and a gift from Amazon Web Services. Additionally, this project made use of time on Tier 2 HPC facility JADE2, funded by the Engineering and Physical Sciences Research Council (EP/T022205/1).

The authors would like to thank Raunak Bhattacharyya, Paul Duckworth, and Matthew Budd for their feedback on an earlier draft of this work.

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
