# OpenReview forum: "RAMBO-RL: Robust Adversarial Model-Based Offline Reinforcement Learning"
_NeurIPS.cc/2022/Conference — NeurIPS 2022 Accept_

### Official Review · Reviewer_xriK · 2022-07-04

**Rating:** 4
**Confidence:** 4
**Soundness:** 1 poor
**Presentation:** 3 good
**Contribution:** 3 good

**Summary:**

The authors tackle the generic Offline RL problem and propose a model-based approach to address it. They introduce a novel algorithm (RAMBO), consisting in adversarially training the model in order to minimize the trained policy performance estimate. The justification of the work builds on a previously published theoretical paper. The empirical validation of the algorithm is limited to MuJoCo-v2 domain where it performs very well, and the AntMaze-v0 where it achieves better than other model-based baselines, but much worse that model-free baselines.

**Questions:**

In addition to addressing the weakness points above, I would be appreciative that the authors answer the following questions:
* Equation 9: why use the *standard* MLE loss instead of the TV loss as the theory prescribes?
* Remark 1: I think that this point is important to make stronger. I fail to see why Problem 1 does not reduce to pessimistic reward modification. My intuition is that the worst possible transition outcome consists in transiting to a state with minimal value, which is equivalent to applying a pessimistic reward modification of this amplitude.

Minor points:
* Line 1: find near-optimal => near-optimality is intractable in general in Offline RL, so this is not the right objective for Offline RL.
* Line 6: achieve conservatism => enforce conservatism.
* I don't like the use of conservatism to refer to any kind of regularisation with respect to the dataset, including pessimism. Generally, conservatism is the regularization of the policy to remain close to that of the behavioural policy, which is generally made in opposition to pessimism which intends to find a lower bound to the value function and does not guarantee to remain close to the behavioural.


**Limitations:**

As I have said before, I am worried that over-stating theoretical findings may lead to improper use of them in the future. Other than that, I do not have any concern with respect to the societal impact of this submission.

**Strengths And Weaknesses:**

Strengths:
* The authors propose a novel algorithm for model-based Offline RL, which makes a lot of sense and is potentially high impact.
* RAMBO performs reasonably well in the experiments in the benchmark.
* The paper is clear overall.

Weaknesses:
* The formalization/theory is not rigorous enough:
  * The theoretical claims are vaguely implying an overstatement of what they are. Near optimality is intractable in Offline RL [Foster2021,Xiao2022] so lines 48-50 makes claims that are too strong. The PAC bounds of [69] are loose in the general case. Similarly, lines 189-190 are much too strong. The constants are not discussed at all, which prevents the reader to understand the true nature of the offered guarantees.
  * Equation 1: what does it mean to have 2 policies in your setting? It is laudable to anchor your work into the literature, but I find it very confusing here, while Problem 1 is much clearer right after.
  * Equation 3: $\widehat{T}_{MLE}$ has been defined, and the concept of it is not well defined in large/continuous environments. I understand that the authors, once again, try to make connection with the literature, but this is again confusing. I would have expected to see here the TV with respect to the dataset, instead, which we learn much later to be what the practical algorithm is doing (section 5.2).
  * Theorem 1: $(1-\gamma)^2$ should be $(1-\gamma)^{-2}$.
  * Proposition 2 is wrongly formalized: $s$ is used both in $V_\phi^\pi(s)$ and in the sampling in the expectation. I have looked into the proof in Appendix A and found that the same imprecision is in the proof. It should actually be instead $z\sim d_\phi^\pi(s)$ where $d_\phi^\pi(s)$ is the state distribution starting from $s$.
* The empirical validation is limited:
  * Some simpler problems (multi-arm bandits, gridworld with finite state-action space) might be useful for a better analysis of what it does better than other model-based Offline RL algorithms.
  * Some harder problems with visual input (Atari for instance) would be useful too as it is where model-based approaches typically fail.

[Foster2021] Offline reinforcement learning: Fundamental barriers for value function approximation, Foster, Dylan J and Krishnamurthy, Akshay and Simchi-Levi, David and Xu, Yunzong (NeurIPS, Offline RL workshop 2021)

[Xiao2022] The Curse of Passive Data Collection in Batch Reinforcement Learning, Xiao, Chenjun and Lee, Ilbin and Dai, Bo and Schuurmans, Dale and Szepesvari, Csaba (AISTATS 2022).

---

> ### Author Response · Authors · 2022-08-02
> **Response to Reviewer xriK**
>
> Thank you for your feedback on our work.
>
> Theoretical claims: The original statements in our paper were paraphrased from the paper which introduced these theoretical results [1]. We have modified lines 48-50 and 189-190 to make the statements less strong as you have suggested.
>
> Equation 1: In our formulation, the model itself can be thought of as the adversary policy in an instantiation of Robust Adversarial RL (RARL). We discuss this in the paper in Remark 2, and this connection between this formulation of offline RL and RARL is one of the contributions we state in the introduction. We have added a reference to Equation 1 in Remark 2 to make this clearer.
>
> Equation 3: In Equation 3, $\hat{T}_{MLE}$ is only considered over the dataset, which is the same as in Section 5.2.
>
> We do not understand the issue you raise about the concept of the MLE model. Please let us know if you feel that this issue has not been resolved, and if so, could you please explain the issue in more depth.
>
> Theorem 1: Thank you for pointing out this typo, it has been corrected.
>
> Proposition 2: Thanks for noting this issue with the notation. In the preliminaries section, we have defined $V^\pi_\phi$ (with no $s)$ to be the expected value under the initial state distribution. We now use this notation on the left hand side of Proposition 2 and in the proof to avoid the issue of $s$ on both sides of the equation.
>
> Empirical validation: We have added additional results on a simple example, specifically to analyse the differences between RAMBO and COMBO, the most similar existing algorithm, in Table 3 and Appendix C.1. These results illustrate that because of the adversarial training used by RAMBO, the value function produced by RAMBO is initially optimistic. Pessimism is introduced into the value function gradually as the model is trained to produce more pessimistic transitions. These new results indicate that this makes policy optimisation less likely to become stuck in local maxima compared to COMBO, which introduces pessimism into the value function at the outset.
>
> Equation 9: Using the TV loss per Equation 8 would have required first training one network to learn $\hat{T}_{MLE}$, and then a second network trained using the TV loss with respect to the first network. Additionally, optimising the TV loss would be more complicated than the MLE. We wished to make our practical implementation as simple as possible to implement, which is why we used the standard MLE loss to train a single model. It is unclear to us that there would have been any practical advantage to using the TV loss approach with two models.
>
> Remark 1: We agree that our approach has a similar ultimate effect to adding reward penalties of a specific magnitude. However, we think it is insightful that appropriate regularisation can be achieved by modifying the MDP model itself, rather than adding penalties. By proposing our approach of adversarially training the dynamics model to minimise the value function, we make a connection between this formulation of offline RL and robust adversarial RL (RARL), as emphasised in Remark 2. Furthermore, the reward penalisation approaches require uncertainty estimation, whereas our approach does not.
>
> Many existing papers which have been impactful share the same approach of penalising the value of out-of-distribution state action pairs, but achieve this regularisation in a different manner. Therefore, we believe our new approach for regularisation can still be a valuable idea to the research community.
>
> Lines 1 & 6: These minor points have been resolved.
>
> Use of the term conservatism: Many existing papers use the term conservatism to refer to approaches which do not constrain the policy to remain close to the behaviour policy, and instead regularise the value function. For example, it is used in the title of the COMBO [2] and CQL [3] papers. Therefore, our usage is consistent with how this term is used in the research community.
>
>
> [1] Uehara, Masatoshi, and Wen Sun. "Pessimistic model-based offline reinforcement learning under partial coverage." ICLR (2021).
>
> [2] Yu, Tianhe, et al. "COMBO: Conservative offline model-based policy optimization." Advances in neural information processing systems 34 (2021): 28954-28967.
>
> [3] Kumar, Aviral, et al. "Conservative Q-learning for offline reinforcement learning." Advances in Neural Information Processing Systems 33 (2020): 1179-1191.

---

> > ### Comment · Reviewer_xriK · 2022-08-07
> > **Not convinced by the rebuttal**
> >
> > Unfortunately, the author-reviewer discussion is in the middle of my holidays, sorry by advance for my telegraphic response.
> >
> > The authors seemingly give what I asked at minimal effort, but things do not really add up. For instance, the experiment does not answer my questions as it has been performed with a single hyperparameter. Simpler experiments are used for deeper analysis with more statistical significance and with hyperparameter sweeps. There are still inconsistencies between the theory and the implementation that have been barely discussed. In general, I feel that the flow of the story of the paper is not well-rounded yet. I'd prefer it to be rejected now to be a good paper at the next conference, rather than a mediocre paper at NeurIPS. I keep my original recommendation: borderline reject.
> >
> > *Furthermore, the reward penalization approaches require uncertainty estimation, whereas our approach does not.* I see that as a bug rather than a feature. Your approach implicitly does uncertainty estimation, meaning that if tomorrow (finally) a reliable uncertainty estimation was to be developed, you would not be able to integrate it out-of-the-box to your algorithm.
> >
> > *Use of the term conservatism* CQL (and COMBO which is basically the same algorithm) has been analyzed as an estimate of the value that is penalized according to the deviation of the target policy wrt the behavioral policy, which in spirit, is conservatism.
> >
> > Eq 3: I mean that in continuous space, generalization comes into play in what $T_{MLE}$ is going to be.

---

> > > ### Author Response · Authors · 2022-08-09
> > > **Response**
> > >
> > > Thanks a lot for getting back to us despite being on holiday.
> > >
> > >
> > >
> > > We have modified the additional experiment so that we now choose the value of the regularisation parameter for COMBO by sweeping over $\beta \in$ {$0.1, 0.25, 0.5, 5.0$}, and selecting the best performance. The best performance for COMBO is 1.39$\pm$0.13 which is obtained using $\beta = 0.25$. For this experiment we used 10 seeds for each configuration (compared to 5 in our main experiments). RAMBO outperforms COMBO for this problem with a statistical significance of $p$=0.02.
> > >
> > >
> > > You say that it is a disadvantage that our approach does not require uncertainty estimation, because if a reliable uncertainty estimation approach is developed in the future, our approach will be less relevant. We think that it is unreasonable to judge the significance of our work based on a hypothetical situation in which an unsolved problem is solved.
> > >
> > >
> > >
> > > Both the COMBO and CQL papers are clear that they use "conservative'' to mean that the algorithms optimise a lower bound on the value function. To quote directly from the CQL paper 'we analyze CQL to show that the policy updates derived in this way are indeed “conservative”, in the sense that each successive policy iterate is optimized against a $\textbf{lower bound on its value.}$' We also address a formulation that optimises a lower bound, so we use the term conservative in the same manner as these previous papers.
> > >
> > > For Equation 3, as we said in our original response $T_{MLE}$, is only considered at points in the dataset. Therefore, it is unclear to us what role the generalisation of $T_{MLE}$ plays in Equation 3.

---

> ### Comment · Area_Chair_diu2 · 2022-08-07
> **Discussion with authors**
>
> Could you please acknowledge that you have read the response to your review? Also, please reply to the authors to indicate how they managed to answer the points raised in your review and how this impacts your score. Finally, make sure that you update your score accordingly.

---

### Official Review · Reviewer_pkCT · 2022-07-06

**Rating:** 6
**Confidence:** 4
**Soundness:** 3 good
**Presentation:** 3 good
**Contribution:** 3 good

**Summary:**

This paper proposes RAMBO, an offline model-based algorithm that formulates the optimization of models and policies as a two-player zero-sum game. Specifically, RAMBO trains models with an extra objective that minimizes the value function to achieve conservative policy optimization. The authors provide a PAC performance guarantee and prove that the learned value function lower bounds the true value function. Experiments demonstrate that RAMBO achieves state-of-the-art performance.

**Questions:**

Please refer to the "Strengths And Weaknesses" part.

**Limitations:**

The authors have addressed parts of the limitations of RAMBO. Nevertheless, I still have concerns about the sensitivity of RAMBO to the hyperparameters.

**Strengths And Weaknesses:**

Originality:

The idea of RAMBO is novel. This paper introduces the idea of RARL into offline model-based RL, which can provide a pessimistic value estimation and alleviate the distribution-shift issue.

Quality:

The theory part is solid.
1. The authors provide a PAC bound that guarantees the accuracy of the learned value function.
2. The author proves that the learned value function is a lower bound of the value function in a real environment.

Nevertheless, the authors may want to provide more details on the algorithm and experiment parts.
1. I can not understand how the gradient is computed according to (5), especially if $s^\prime$ is a terminal state.
2. Appendix shows that RAMBO is sensitive to hyperparameters, especially in Hopper and Walker2D. I wonder whether the gradient in (5) has high variance if $s^\prime$ is a terminal state, and then the optimization of models becomes unstable.
3. I wonder whether the model training phase in RAMBO leads to high computational cost. The authors may want to compare RAMBO and COMBO in wall-clock time.
4. I wonder whether RAMBO learns a lower bound of the true value function in authors' empirical studies. If the authors can provide analyses similar to those in TD3 [1] and REDQ [2], the authors' claims would be better supported.

Clarity:

This paper is well written and easy to follow.

Significance:

This paper provides a new idea for learning a pessimistic value function. However, the performance improvement is insignificant, as COMPO outperforms RAMBO^OFF as shown in Appendix.

[1] Fujimoto S, Hoof H, Meger D. Addressing function approximation error in actor-critic methods[C]//International conference on machine learning. PMLR, 2018: 1587-1596.

[2]Chen X, Wang C, Zhou Z, et al. Randomized Ensembled Double Q-Learning: Learning Fast Without a Model[C]//International Conference on Learning Representations. 2020.

---

> ### Author Response · Authors · 2022-08-02
> **Response to Reviewer pkCT**
>
> Thank you for your feedback on our work.
>
> 1).
> In Equation 5, if the reward plus the value at the successor state, $r + \gamma V^\pi_\phi(s’)$, are lower than what was expected compared to $Q^\pi_\phi(s, a)$ then the parameters of the model, $\phi$, will be updated to make receiving reward $r$ and transitioning to $s’$ more likely after executing $(s, a)$. Conversely, if the reward plus value at the successor state are higher than expected the model parameters will be updated to make receiving that reward and successor state less likely. This is analogous to policy gradient algorithms, except that we modify the distribution over rewards and successor states, rather than over actions.
>
> If $s’$ were to be a terminal state, then $V^\pi_\phi(s’)$ would be low as no future reward can be gained from $s’$. Because the value of reaching $s’$ is low, the adversarial update according to Equation 5 will make transitions to $s’$ more likely.
>
> 2).
> Compared to other offline model-based RL algorithms, RAMBO does not appear to be particularly sensitive to hyperparameters. For example, in the appendix of the COMBO paper, significant variation in performance is also reported for different hyperparameter values.
>
> We also do not think that RAMBO is particularly difficult to tune. In our experiments, we only modified two hyperparameters across a total of three different configurations. For COMBO, six different hyperparameters were modified between datasets (rollout length, learning rates, conservative coefficient, regularisation distribution, rollout policy, ratio of real data).
>
> 3).
> In the experiments section, we report a training time of 24-30 hours for RAMBO. COMBO reports a computation time of one day. While RAMBO has to additionally adversarially train the transition model, COMBO requires optimising the additional regularisation term in the value function update. It appears the result is comparable computation time.
>
> 4).
> Thank you for this suggestion. However, unfortunately we did not have time to run this analysis during the rebuttal period.

---

> > ### Comment · Reviewer_pkCT · 2022-08-04
> > **Thank the authors for the response**
> >
> > Thank the authors for the response. I am still confused about Equation (5).
> > 1. According to the authors' response, the gradient is computed by $V_\phi^\pi(s^\prime)$. Is the $V_\phi^\pi(s^\prime)$ an additional network?
> > 2. I wonder whether the terminal condition makes the $V_\phi^\pi(s^\prime)$ discontinuous and thus leads to a high variance of the gradient. According to Table 1 and Table 6, the performance of RAMBO is relatively poor in Hopper and Walker, and the value function even diverges in these two environments without tuning the hyperparameters. The authors may want to provide more discussion/results to demonstrate the effectiveness of RAMBO to deal with terminal states.
> > 3. Moreover, we wonder whether the data $(s, a, r, s^\prime)$ is included in the training set if $s$ is a terminal state. In popular implementations, this data is not included in the training set. However, in this paper, we need an accurate estimate of $V^\pi(s^\prime)$, even if $s^\prime$ is a terminal state.
> >
> > Moreover, I wonder whether the hyperparameters of RAMBO are tuned with online evaluation. The authors may want to provide a method that tunes these hyperparameters offline, as online evaluation may be risky in some real-world applications.

---

> > > ### Author Response · Authors · 2022-08-04
> > > **Response to Reviewer pkCT**
> > >
> > > Thank you for responding to us so quickly.
> > >
> > > Our apologies, we misunderstood your original question regarding terminal states. Thank you for clarifying your concern. We hope that we have addressed the issues you have raised in our response below, but please do let us know if you have further questions.
> > >
> > >
> > >
> > > $V^\pi_\phi$:
> > >
> > > $V^\pi_\phi$ is the value function learned by the actor-critic RL algorithm used to optimise the policy. It is a separate network, defined by different parameters ($\theta$, lets say). In the paper, we use the subscript of the value function, $V^\pi_M$, to indicate that the value function is defined for MDP $M$. So,  in the case of $V^\pi_\phi$ the subscript $\phi$ indicates that this is the value function trained in the MDP model defined by parameters $\phi$ (denoted by $\widehat{T}_{\phi}$ in the paper). To reiterate, $V^\pi_\phi$ itself is a separate network defined by different parameters, $\theta$.
> > >
> > >
> > > $\textbf{Terminal states}$:
> > >
> > > We originally misunderstood your question concerning terminal states. We clarify how terminal states are handled in our implementation here.
> > >
> > > In our implementation, we assume that we have access to the termination function of the MDP, which we refer to here as $terminal(s)$. This is a standard assumption in model-based RL. In our code, we compute $r + \gamma V^\pi_\phi(s')$ as required by Equation 5 as: $r + \gamma \cdot (\textnormal{not}\ terminal(s')) \cdot V^\pi_\phi(s') $. This is the same method used to compute the Bellman error in most RL implementations, and is implemented in line 890 of rambo/algorithms/rambo.py of our code in the supplementary material.
> > >
> > > Thus, if $s'$ is terminal, our implementation treats $s'$ as if it has zero value due to the use of the termination function. Therefore, it does not matter if the estimate of $V^\pi_\phi(s')$ is inaccurate at terminal states, as the value utilised at terminal successor states is always set to zero.
> > >
> > > Because terminal states are always treated as having zero value, we do not think that the handling of terminal states leads to instability in our algorithm.
> > >
> > >
> > > $\textbf{Sensitivity to hyperparameters}$:
> > >
> > > As you have pointed out, the performance of our algorithm does vary significantly with untuned hyperparameters, specifically the rollout length.
> > > However, we think that including these results is actually a strong point of our submission, as it provides transparency as to how well our algorithm performs for a variety of hyperparameters.
> > > Existing commonly cited model-based offline RL papers (COMBO, MOPO, MOReL) only provide results for tuned values of the rollout length.
> > > Therefore, it is unclear whether our algorithm is more sensitive to the rollout length than other existing algorithms.
> > >
> > > We do not think that the inclusion of these additional results for untuned hyperparameters should be considered a negative aspect of our submission.
> > > Penalising a submission based on additional results for untuned hyperparameters encourages authors to only present cherry-picked results in their papers.
> > > We think that including a wider range of results, including for situations in which the algorithm does not perform as well, makes it easier for researchers to assess the potential limitations of a given approach.
> > >
> > >
> > > $\textbf{Offline hyperparameter tuning}$
> > >
> > > In Appendix C.3, we provide results where the hyperparameters were selected using a heuristic that is evaluated offline. To generate the results in Appendix C.3, we rerun the algorithm after selecting the hyperparameters offline. These results demonstrate that we can still achieve solid performance with RAMBO using only offline hyperparameter tuning.
> > >
> > > The results presented in the main body of the paper use online hyperparameter tuning by comparing three possible configurations of the hyperparameters (details in Appendix B.4).

---

> > > > ### Comment · Reviewer_pkCT · 2022-08-08
> > > > **Thank you for the response**
> > > >
> > > > Thank you for the response. I have understood how RAMBO deals with the terminal states.  I will increase my score.

---

### Official Review · Reviewer_dsJ3 · 2022-07-08

**Rating:** 6
**Confidence:** 4
**Soundness:** 3 good
**Presentation:** 4 excellent
**Contribution:** 2 fair

**Summary:**

To remedy with the extrapolation error in offline setting, this paper formulates the offline RL as an adversarial process between the agent and the environment. The environment is modeled as a neural network and trained with both MLE objective and conservative objective which is essentially minimizing the state value of the agent. Previous research [1] proved that this formulation enjoys a PAC performance guarantee, and the experiments do demonstrate some performance improvement on certain D4RL tasks.

[1] Masatoshi Uehara and Wen Sun. Pessimistic Model-Based Offline Reinforcement Learning under Partial Coverage. ICLR 2022.

**Questions:**

Just one question about COMBO baseline. In the original paper of COMBO, it was not clarified which version of the dataset (v0 or v2) was used in the experiment, so I suppose COMBO used v0 datasets as CQL did. So did you re-implement COMBO and tested it with v2 dataset?

Also, it is interesting that the performance of COMBO reported in this paper is precisely identical to the original paper, except for walker2d-medium-expert, hopper-medium-replay, hopper-medium and walker2d-medium. So would you elaborate more on how did you obtain the result of COMBO?

Please correct me if there is any mis-understanding.


**Limitations:**

The authors do mention some limitations of their work at the conclusion section, such as the computational cost. However I don't think this is a major concern.


**Strengths And Weaknesses:**

Overall I enjoyed reading this paper.
+ This paper is well organized and comprehensive in presentation. The authors manage to convey their idea in concise but informative language.
+ The main argument that adversarial training can relieve value overestimation and model defect is supported by an illustrative example in the appendix. This is a persuasive argument to justify the effectiveness of the proposed method.
+ The authors provide the detail about their hyper-parameter configuration and strategy in the appendix. I think the strategy of deciding on RAMBO's hyper-parameters is acceptable, for no excessive online fine-tuning is involved in training stage.

Some weaknesses:
+ The baselines, no matter model-based ones or model-free ones, seem to be old at this time. For model-free ones, there are [2] and [3]; for model-based algorithms, there is [4] as far as I can recall.
+ The analysis of RAMBO is too brief. From table 2 I can tell that RAMBO does better on medium-replay datasets while (comparatively) worse on medium-expert datasets, but the authors did not provide in-depth analysis of this phenomenon. More discussions, such as whether this phenomenon is related to adversarial training and how, is expected.

[2] Gaon An, Seungyong Moon, Jang-Hyun Kim, Hyun Oh Song. Uncertainty-Based Offline Reinforcement Learning with Diversified Q-Ensemble. NeurIPS 2021.

[3] Ilya Kostrikov, Rob Fergus, Jonathan Tompson, Ofir Nachum. Offline Reinforcement Learning with Fisher Divergence Critic Regularization. ICML 2021.

[4] Yijun Yang, Jing Jiang, Tianyi Zhou, Jie Ma, Yuhui Shi. Pareto Policy Pool for Model-based Offline Reinforcement Learning. ICLR 2022.

---

> ### Author Response · Authors · 2022-08-02
> **Response to Reviewer dsJ3**
>
> Thank you for your feedback on our work.
>
> Because the original COMBO paper used the -v0 datasets, we used the results for COMBO on the -v2 datasets reported by [1]. We are not sure why these results differ from the original COMBO paper for only some of the datasets, and have contacted the authors of [1] to seek clarification as to how these results were generated.
>
> We have added further analysis of RAMBO, specifically in comparison to COMBO to the experiments section (Table 3) and Appendix C.1. These results illustrate that the value function for RAMBO is initially optimistic. Because of the adversarial training, pessimism is introduced into the value function gradually by RAMBO as the model is trained to generate more pessimistic transitions. These results indicate that this makes the policy optimisation less likely to become stuck in local maxima compared to COMBO (and other existing algorithms), which introduce pessimism into the value function at the outset. We believe that this demonstrates how RAMBO differs from existing approaches, and that gradually increasing the level of pessimism can be a useful modification to existing offline RL algorithms.
>
> For the baselines in the experiments section, we opted to include algorithms that are commonly compared against and that most readers will be familiar with. We also chose to compare against algorithms for which results on AntMaze-v0 were available. We have added references to the papers you have mentioned to the related work section.
>
> [1] Jianhao Wang, Wenzhe Li, Haozhe Jiang, Guangxiang Zhu, Siyuan Li, and Chongjie Zhang. Offline reinforcement learning with reverse model-based imagination. Advances in Neural Information Processing Systems, 34, 2021.

---

> > ### Comment · Reviewer_dsJ3 · 2022-08-07
> > **Thanks for your explanation.**
> >
> > I have read the analysis in Appendix C.1 and I think the results do take a step further in revealing the mechanism of adversarial model training. I have one more concern about the Appendix C.1 experiments, because the degree of the pessimism on Q values is also affected by certain hyper-parameters. I noticed that you select the lowest value of \beta from the original paper, but it would be more persuative to include results where \beta is smaller (thus reduce the degree of pessimism) while COMBO still achieves comparable performance on this toy environment.

---

> > > ### Author Response · Authors · 2022-08-09
> > > **The paper has been updated**
> > >
> > > Thanks for this suggestion.
> > >
> > > We have changed the result in the paper (and in Appendix C.1) so that we now select the regularisation hyperparameter for COMBO by sweeping over $\beta \in$ {$0.1, 0.25, 0.5, 5.0$} and choosing the best performance. This includes values lower than those used in the original COMBO paper (i.e. < 0.5).
> > >
> > > The best performance obtained by COMBO was 1.39$\pm$0.13, which was obtained using $\beta = 0.25$. This is worse than the performance obtained by RAMBO for this problem (with signficance of $p$ = 0.02).

---

### Official Review · Reviewer_j6Y8 · 2022-07-11

**Rating:** 6
**Confidence:** 4
**Soundness:** 3 good
**Presentation:** 3 good
**Contribution:** 2 fair

**Summary:**

The authors propose RAMBO to offline RL which imposes conservatism by adversarially modifying the transition dynamics of a learned model. RAMBO is built on a formulation for which PAC bounds are guaranteed. The experiments are conducted in the D4RL benchmark and several SOTA baselines are taken for comparison.

**Questions:**

The article is well-written and soundness to me. I have no further questions.

**Limitations:**


As mentioned above, I think further comparison should be considered to clarify the advantages/disadvantages/scopes of the two regularizations.If it is done and we can find more insights in the two regularizations after that,  I will consider increasing the score

**Strengths And Weaknesses:**


Strengths
1. The article is well-written and easy to follow.
2. The solution is simple to implement and sounds reasonable.


Weaknesses

1. RAMBO is similar to COMBO: COMBO learns a conservative Q through regularization based on the policy while RAMBO reduces unreasonable V through regularization (Adversarial training) based on the model. The results in Table 1 also show the similarity in performance. It is OK to give a novel regularization from other perspectives. But I think further comparison should be considered to clarify the advantages/disadvantages/scopes of the two regularizations.

---

> ### Author Response · Authors · 2022-08-02
> **Response to Reviewer j6Y8**
>
> Thank you for your feedback on our work.
>
> We have added additional results comparing RAMBO and COMBO in the experiments section (Table 3) and Appendix C.1. These results illustrate a key difference between RAMBO and COMBO. RAMBO introduces pessimism gradually as the adversary is trained to generate more pessimistic transitions. For COMBO, pessimism is present in the value function throughout training as it is part of the value function update.
>
> In the example we have added, there are several distinct regions of the action space which are covered by the dataset. Because the value function produced by RAMBO is initially optimistic, and pessimism is introduced gradually, there is less chance of the policy learned by RAMBO becoming stuck in poor local maxima in this example. This is a natural characteristic of RAMBO. For COMBO, we observe that the value function contains local maxima throughout training, and it is more likely that the final policy finds a local maximum. We believe that this illustrates a crucial difference between RAMBO and COMBO, and that gradually increasing the level of pessimism can be a useful modification to existing offline RL algorithms.

---

> > ### Comment · Reviewer_j6Y8 · 2022-08-06
> > **Response**
> >
> > Thanks for your response. I have read the results in Appendix C.1. I think the results are interesting and the difference in Fig. 2 and Fig. 3 strongly demonstrate the difference between the two regularizer mechanisms.
> >
> > Now I have a further question about the problem. The authors state that "RAMBO consistently performs better than COMBO for this problem " (line 784-785), but in Medium-Expert (MuJoCo) and AntMaze, RAMBO is worse than COMBO in general. Can the authors give further explanation on this?
> >
> > On the other hand, did you conduct the experiments in Appendix C.1 on different tasks? Does the difference generally exist?

---

> > > ### Author Response · Authors · 2022-08-06
> > > **Response to Reviewer j6Y8**
> > >
> > > Thanks for getting back to us. We are glad that you appreciate the additional results we added in Figures 2 and 3.
> > >
> > >
> > >
> > > We think it is difficult to definitively conclude what causes the differences between the algorithms on the more complex domains. However, upon reviewing the results for the medium-expert datasets in more detail, we observe that for Walker2D medium-expert there is initially some instability in the Q-values during training on some runs, which we do not observe for other datasets. This likely happens because as illustrated in Appendix C.1, regularisation is introduced by RAMBO gradually, so there is a tendency for values to be over-estimated early in training. This unstable training for some runs results in a less performant policy on these runs. This is reflected by the fact that the variation in the performance between seeds is very high for Walker2d medium-expert compared to the other datasets. We are unsure why we observe this issue on Walker2D medium-expert specifically. We did not observe this phenomenon for the Hopper and Halfcheetah medium-expert datasets, but RAMBO still performs reasonably well on these problems.
> > >
> > >
> > >
> > > For Antmaze, we did not observe the same instability during training so this does not appear to explain the performance for Antmaze. One possible explanation is that because the Antmaze domain is higher-dimensional, it requires more training to appropriately train the adversarial model. However, we did try running the Antmaze datasets for more iterations, and this did not seem to improve performance. We do wish to point out that RAMBO does perform better than COMBO for some of the Antmaze datasets, and achieves a scores greater than 0 across most of the datasets, unlike COMBO.
> > >
> > >
> > >
> > > While generating the results in Appendix C.1, we did test some other slight variations of the one-dimensional "Single Transition" problem. The overall tendency was that the policy produced by COMBO was more likely to become stuck in a local minima. However, we did find that for some variations of the problem (particularly those in which there was one action that was distinctly better than the rest), both algorithms had no trouble finding the globally optimal action.

---

> > > > ### Comment · Reviewer_j6Y8 · 2022-08-06
> > > > **Response**
> > > >
> > > > Thanks for your response. I will increase my score.

---

### Meta-Review · Area_Chair_diu2 · 2022-08-22

**Recommendation:** Accept
**Confidence:** Certain

**Metareview:**

This paper introduces the idea of Robust Adversarial RL for offline model-based RL, which could have a high impact. It is well organized and the writing is very comprehensive; the authors manage to convey their idea in concise but informative language. The proposed RAMBO approach performs reasonably well in the presented experiments, although it was pointed out that the paper would benefit from more scenarios showing the necessity of RAMBO compared with the current baseline (COMBO). Questions and issues related to the theory that were raised during the reviewing process have been addressed in the rebuttal.



**Award:**

No

---

### Decision · Program_Chairs · 2022-09-14

Accept